# A Comprehensive In Silico Study of New Metabolites from *Heteroxenia fuscescens* with SARS-CoV-2 Inhibitory Activity

**DOI:** 10.3390/molecules27217369

**Published:** 2022-10-29

**Authors:** Fahd M. Abdelkarem, Alaa M. Nafady, Ahmed E. Allam, Mahmoud A. H. Mostafa, Rwaida A. Al Haidari, Heba Ali Hassan, Magdi E. A. Zaki, Hamdy K. Assaf, Mohamed R. Kamel, Sabry A. H. Zidan, Ahmed M. Sayed, Kuniyoshi Shimizu

**Affiliations:** 1Department of Pharmacognosy, Faculty of Pharmacy, Al-Azhar University, Assiut 71524, Egypt; 2Department of Pharmacognosy and Pharmaceutical Chemistry, College of Pharmacy, Taibah University, Al Madinah Al Munawarah 41477, Saudi Arabia; 3Department of Pharmacognosy, Faculty of Pharmacy, Sohag University, Sohag 82524, Egypt; 4Department of Chemistry, Faculty of Science, Imam Mohammad Ibn Saud Islamic University (IMSIU), Riyadh 11623, Saudi Arabia; 5Department of Pharmacognosy, Faculty of Pharmacy, Nahda University, Beni-Suef 62513, Egypt; 6Department of Agro-Environmental Sciences, Graduate School of Bioresource and Bioenvironmental Sciences, Kyushu University, Fukuoka 819-0395, Japan

**Keywords:** sterol, virus, sesquiterpene, soft coral, in silico, molecular dynamic simulation, docking

## Abstract

Chemical investigation of the total extract of the Egyptian soft coral *Heteroxenia fuscescens*, led to the isolation of eight compounds, including two new metabolites, sesquiterpene fusceterpene A (**1**) and a sterol fuscesterol A (**4**), along with six known compounds. The structures of **1**–**8** were elucidated via intensive studies of their 1D, 2D-NMR, and HR-MS analyses, as well as a comparison of their spectral data with those mentioned in the literature. Subsequent comprehensive in-silico-based investigations against almost all viral proteins, including those of the new variants, e.g., Omicron, revealed the most probable target for these isolated compounds, which was found to be M^pro^. Additionally, the dynamic modes of interaction of the putatively active compounds were highlighted, depending on 50-ns-long MDS. In conclusion, the structural information provided in the current investigation highlights the antiviral potential of *H. fuscescens* metabolites with 3*β*,5*α*,6*β*-trihydroxy steroids with different nuclei against SARS-CoV-2, including newly widespread variants.

## 1. Introduction

The global pandemic caused by coronavirus 2 (SARS-CoV-2) has taken many lives [1,2] from 2019 until the present. Several variants and outbreaks are still present in countries such as China and Korea [3].

Coronaviruses are a family of viruses that can cause a wide array of enteric, hepatic, neurological, and respiratory diseases [1]. The four subfamilies of coronavirus are α, β, γ, and δ, and are sorted based on genotype and serotype data [4]. Six species of coronaviruses cause human diseases. The most pathogenic and fatal species are Severe Acute Respiratory Syndrome Coronavirus (SARS-CoV) and Middle East Respiratory Syndrome Coronavirus (MERS-CoV). SARS-CoV-2 is the seventh type, which infects humans and is related to the β lineage of the beta-coronaviruses, which are known to cause severe disease and fatalities [4]. Based on genomic analysis, SARS-CoV-2 belongs to two bat-derived SARS-like coronaviruses with 96% similarity, whereas its similarity to SARS-CoV is 79% [5]. The lung is the primary site and the most affected organ of SARS-CoV-2 infection. The clinical manifestations range from asymptomatic to severe respiratory disease. The most common symptoms of infection are a loss of smell and taste, fever, headache, shortness of breath, cough, muscle aches, and tiredness [6].

The coronaviral genome encodes four structural proteins, namely the envelope (E) protein, membrane (M) protein, nucleocapsid (N) protein, and spike (S) protein [5]. Most vaccine strategies have been focused on the spike protein, as it plays an essential role in viral entry to the host cell [5]. The E protein and M protein participate in the production and release of virus-like particles, while the N protein is a critical for viral replication and genome packaging [7,8]. During virus infection, the main proteases (M^pro^) and cyclin-dependent kinases (CDKs) play a crucial role in the regulation and progression of cell division, which is essential in replication and viral transmissibility [9,10]. The main protease (M^pro^) is essential for processing the polyproteins that are translated from the viral RNA, and SARS-CoV-2 modifies the CDK signaling pathway and regulates the cell division cycle in a way that leads to the enhancement of viral replication [11,12]. Consequently, M^Pro^ and CDK inhibitors are considered attractive targets for drugs that are used to treat the SARS-CoV-2 virus [13].

To speed up the treatment process, several antiviral drugs have been re-evaluated as an emerging antiviral option in the SARS-CoV-2 infection. Favipiravir has shown promising results in clinical studies in Japan, Russia, and China. Additionally, the oral antiviral drug nirmatrelvir/ritonavir (Paxlovid) has been approved by the FDA for emergency use [14].

Naturally occurring metabolites have proven to be promising sources of drug leads and structural motifs, based on the diversity of their structures and biological activities [15,16,17]. Several compounds, such as sesquiterpenes, diterpenes, steroids, alkaloids, and lipid derivatives, were isolated from soft corals [18,19,20]. Soft corals belonging to the family Xeniidea (Alcyonacea) are distributed throughout the oceans and seas [21,22], and the two genera that have been subjected to detailed chemical and biological investigation in the Red Sea are *Xenia* and *Heteroxenia* [23,24]. The genus *Heteroxenia* is distributed along the Egyptian Red Sea’s coast, with two common species, *Heteroxenia fuscescens* and *Heteroxenia ghardaqensis* [21]. Several secondary metabolites, such as steroids (with a high degree of oxygenation), sesquiterpenoids, ceramides, and glycerol derivatives, were isolated from both species. Many of these metabolites showed diverse biological activities, such as anticancer, anti-inflammatory, anti-diabetic, and analgesic activities [24,25,26,27,28,29,30].

Based on the above data, as well as the urgent necessity of developing and searching for natural drugs to treat the SARS-CoV-2 virus, the isolation of the secondary metabolites of the soft coral *Heteroxenia fuscescens*, followed by the in silico prediction of isolated compounds as potential inhibitors of M^pro^ and Cyclin-G-associated kinase (GAK) and a promising scaffold using molecular docking and modeling analysis, were carried out.

## 2. Results

### 2.1. Identification of the Isolated Compounds

The crude extract of *H. fuscescens* was partitioned into fractions and isolated using several chromatographic techniques to yield eight compounds (**1**–**8**) (Figure 1). The structures of the known compounds were elucidated to be 3β,6β-dihydroxy-10β-hydroperoxycadin-4,8-diene, (**2**) [25], gorgost-5-(E)-ene-3-β-ol (**3**) [31,32], gorgost-3β,5α,6β,11α-tetrol (**5**) [33,34], 11α-acetoxy-gorgost-3β,5α,6β-triol (**6**) [30,35], 14α,15α-epoxy-23,24-dimethylcholesta,3β,5α,6β,11α,20α-pentol (**7**) [25], and 3β,5α,6β-trihydroxyandrosta-17-one (**8**) [25], based on comparisons of their 1D (^1^H and ^13^C) NMR spectroscopic data against the data in the literature (Appendix A).

Compound **1** was obtained as a colorless oil. The positive HR-FAB-MS showed a [M+H]^+^ at *m/z* 235.1701 (calcd for C_15_H_23_O_2_, 235.1698) and a base peak at [M-H_2_O+H]^+^ *m/z* 217.1605 (calcd for C_15_H_21_O, 217.1592); together with NMR spectroscopic data, the molecular formula was established to be C_15_H_22_O_2_, with five degrees of unsaturation (Appendix A).

The combined analysis of the ^13^C-NMR and HSQC spectra displayed 15 signals (Appendix A), which were categorized as four quaternary carbons (including three aromatic carbons at *δ_C_* 139.5, 137.7, and 136.8, and one oxygenated at *δ_C_* 74.5), six methines (including three aromatic carbons at *δ_C_* 125.4, 127.5, and 128.8, and one oxygenated carbon at *δ_C_* 73.8), one methylene at *δ_C_* 27.3, and four methyls (including two quaternaries at *δ_C_* 21.2, and 24.3, and two tertiaries at *δ_C_* 18.0 and 21.6). The ^1^H-NMR spectrum (Appendix A) indicated the presence of three aromatic protons, including two ortho-coupling protons at *δ_H_* 7.5 (d, *J* = 7.9 Hz, H-2) and *δ_H_* 7.1 (d, *J* = 7.9 Hz, H-3), and one singlet proton at *δ_H_* 7.0 (s, H-5). One oxygenated proton at *δ_H_* 4.0 (dd, *J* = 11.7, 3.9 Hz, H-9), two quaternary methyls at [*δ_H_* 1.4 (s, H_3_-14), 2.3 (s, H_3_-15)], and two tertiary methyls at [*δ_H_* 0.78 (d, *J* = 7.3 Hz, H_3_-13), *δ_H_* 1.06 (d, *J* = 7.1 Hz, H_3_-12)]. Three double bonds accounted for three degrees of unsaturation; consequently, the remaining degrees were due to the bicyclic structure of **1**. From these data, compound **1** is assumed to be a bicyclic calamenene-type sesquiterpene [36,37] with two oxygenated carbons. The assignment of all carbons and protons, as well as the structure of compound **1**, was elucidated based on the extensive analysis of 2D NMR (^1^H-^1^H COSY, HSQC, and HMBC).

The HMBC spectrum (Figure 2 and Appendix A) clearly illustrated the attachment of H_3_-15 at C-4 via the presence of a correlation between the two ortho-coupling protons H-2 and H-3, and the observed correlations between H_3_-15 and C-3, C-4, and C-5. The second quaternary methyl H_3_-14 was attached to the oxygenated carbon C-10 based on the existence of cross-peaks between H_3_-14 and C-1, C-9, and C-10. The second oxygenated carbon was C-9, based on the downfield shift of H-9 *δ_H_* 4.0 (dd, *J* = 11.7, 3.9 Hz, H-9), while the isopropyl moiety was attached to C-7 based on the HMBC correlations between H_3_-12 and C-7, C-11, and C-13; and between H_3_-13 and C-7, C-11, and C-12, respectively. The ^1^H-^1^H COSY spectrum (Appendix A) illustrated the presence of two discrete spins corresponding to H-2/H-3 in the aromatic ring and H-9/H_2_-8, H_2_-8/H-7, H-7/H-11, H-11/H_3_-12, and H-11/H_3_-13, which confirmed the overall structure of compound **1**.

The relative configuration of compound **1** (7*R**,9*S**,10*R**)-*cis* calamenene was deduced from the investigation via 1D and 2D NMR analyses (Figure 3, Appendix A), and via a comparison with the previously reported data of related compounds [38,39]. In particular, the main NOE correlations were observed between H-9 and H_3_-14. Additionally, the absence of an NOE correlation between H-9 or H_3_-14 and H-7 indicated that H-9 and H_3_-14 are in one direction, and H-7 is in the other direction. By comparing the chemical shifts of H-7 and C-7 (*δ_H_* 2.84 m; δ_c_ 41.5) of compound **1** with relative metabolites as *cis*-muurola-4(14),5-diene with the ^13^C-NMR chemical shift of C-7 (δ_c_ 51.7) and 3-hydroxy calamenene with the ^13^C-NMR chemical shift of C-7 (δ_c_ 43.9) (Table 1) [38,39], we can conclude that the OH at C-9 and C-10 were *β*-oriented, while an isopropyl group at C-7 was α-oriented. From the above data, compound **1** is new, identified as (7*R**,9*S**,10*R**) 9,10-dihydroxycalamenene and named fusceterpene A.

Fuscesterol A **4** was isolated as a white powder. The HR-ESI-MS of compound **4** exhibited a [M+Na]^+^ peak at *m/z* 527.3673 corresponding to the molecular formula C_31_H_52_O_5_ and implying six degrees of unsaturation (Figure 1). These observations were in agreement with 31 signals in the ^13^C-NMR spectrum (Table 2 and Appendix A). They were clarified via an HSQC experiment (Appendix A) into six quaternary carbons (including one carbonyl at *δ_C_* 172.8, 2 oxygenated at *δ_C_* 87.7 and 76.6, and one olefinic at *δ_C_* 152.3), eight methines (including two oxygenated at *δ_C_* 76.4, 73.0), 10 methylenes (including one terminal olefinic at *δ_C_* 112.0), and seven methyls. From the above data, compound **4** was a steroid in nature [40,41]. Four degrees of unsaturation were accounted for by the tetracyclic structure, one for vinylic methylene and one for carbonyl. The above data suggested that compound **4** was an acetoxylated dimethyl cholestane-type steroid with a terminal double bond on the side chain [42,43,44].

^1^H-NMR data (Appendix A) showed signals for seven methyls, including four doublets at *δ_H_* 0.93 (3H, *J* = 6.6 Hz), 0.89 (3H, *J* = 6.6 Hz), 0.80 (3H, *J* = 6.6 Hz), and 0.78 (3H, *J* = 6.6 Hz), which were assigned to H_3_-26, H_3_-27, H_3_-28, and H_3_-29, and two singlets at *δ_H_* 0.65 and 1.18, assigned to H-18 and H-19, respectively. Two singlet signals at *δ_H_* 5.14 s and 4.89 s were assigned to vinylic methylene at C-21. The location of the double bond was confirmed via the presence of cross-peaks between H_2_-21 and C-17, C-20, and C-22 in the HMBC spectrum (Appendix A). The presence of a hydroxylated group at C-17 (87.7) neighboring the vinylic methylene was confirmed from the HMBC correlation between H_2_-21, H_3_-18, and H_2_-16 with C-17. The downfield shift of the oxygenated methines at *δ_H_* 5.17 (m) at C-3 was due to the attachment of an acetate moiety. The attachment was confirmed via the presence of cross-peaks between H-3 and C-2, C-4, and C-1′, respectively. The key ^1^H-^1^H COSY correlations were observed between H-2/H-3, H-3/H-4, H-7/H-8, H-14/H-15, and H-15/H-16 (Appendix A), which confirm the structure of compound **4**.

The relative stereochemistry of compound **4** was deduced based on an NOE correlation in the NOESY spectrum (Appendix A) and comparing the ^1^H and ^13^C-NMR with the relative metabolites [42,44]. The key NOESY H-3/H-4a (*δ_H_* 1.58), H-4a/H-6, and H_3_-19/H-4b (*δ_H_* 2.16) indicated that OAc-3 and OH-6 were *β*-oriented. The *α*-orientation of the OH at C-5 and C-17 was clarified by comparing the ^13^C-NMR of C-5 and C-17 (*δ_C_* 76.6 and δc 87.7), respectively, with relative metabolites as (23*R*) methyl-ergosta-20-ene-3*β*,5*α*,6*β*,17*α*-tetrol (*δ_C_* 76.9 & δc 87.7), gorgostan-3*β*,5*α*,6*β*-triol-11*α*-acetate (C-5 *δ_C_* 76.5), and klyflaccisteroid A (C-17 *δ_C_* 86.7) [30,42,44]. Additionally, the relative configuration of the side chain was confirmed from the good agreement with the ^1^H and ^13^C-NMR data (23*R*, 24*R*)-23,24-dimethyl-cholest-20-ene side chain. Based on the above finding, compound **4** was identified as (23*R**,24*R**)-3*β*-acetoxy-5*α*,6*β*,17*α*-trihydroxy-23,24-dimethylcholesta-20-ene and named fuscesterol A.

### 2.2. Molecular Docking and Molecular Dynamic Simulation Results

To putatively determine the most probable molecular target for compounds **1**–**8**, we virtually screened their structures against all currently available SARS-CoV-2 proteins that have been reported to be relevant to the viral life cycle or viral pathogenesis (https://swissmodel.expasy.org/repository/species/2697049; https://www.genome.jp/kegg-bin/show_pathway?hsa05171+H02398 (accessed on 1 September 2022); Appendix A). Both M^pro^ and Cyclin-G-associated kinase (GAK) were the only proteins that were found to have considerable docking scores (<−8.0 kcal/mol) with compounds **4**, **7**, and **8** (for M^pro^) and compounds **3**–**5** and **8** (for GAK).

The subsequent MDS validation experiments (50 ns long) revealed that the structures of compounds **4**–**7** and **8** were not good binders to GAK, because they were significantly unstable over the course of MDS, with an average RMSD ranging from 6Å to > 10 Å. Moreover, their calculated absolute binding free energy values (ΔG_binding_) indicated a low affinity toward GAK’s binding site, and ranged from −2.5 to −3.2 kcal/mol.

In contrast, the structures of compounds **5**, **7**, and **8** have achieved stable binding over the course of 50 ns MDS runs, with average RMSDs ranging from 1.8 Å to 2.2 Å, and with ΔG_binding_ values of −8.6, −8.8, and −8.1 kcal/mol, respectively. Interestingly, the three steroids that shared the stable binding had a characteristic hydroxylation pattern at positions 3, 5, and 6 in rings A and B. In addition, we have reported that the triterpene class of compounds, which is structurally related to compounds **5**, **7**, and **8**, is a promising scaffold to develop new M^pro^ inhibitors [45,46].

To investigate the binding mode of each compound inside the M^pro^’s active site, the most populated binding poses for each compound structure was extracted from their 50-ns-long MDS trajectories.

As shown in Figure 4, the binding modes of the investigated structures were different from each other. The structure of compound **5** established three stable H-bonds with THR-25, ASP-187, and GLN-189, in addition to two hydrophobic interactions with LEU-27 and MET-49. The structure of compound **7** also established three H-bonds, but with ASN-142, GLU-166, and GLN-189. Additionally, it interacted hydrophobically with LEU-27 and MET-49. Finally, the structure of compound 8 was able to form four stable H-bonds with VAL-42, TYR-54, ASN-142, and GLN-189, together with two hydrophobic interactions with LEU-27 and MET-49.

## 3. Discussion

In the previous literature, several bioactive metabolites were isolated from soft corals, which have the ability to inhibit the SARS-CoV-2 M^pro^, as depresosterol, lopophytosterol, and cholest-5-ene-3*β*,7*β*-diol [47]. These compounds form stable hydrogen bonding through a polyhydroxy group in their structure with the amino acid in the active pocket of the SARS M^pro^ [47]. In the genus *Heteroxenia* (family Xeniidae), several polyoxygenated steroids were isolated and showed diverse biological activity [24,25,27,28,42].

From the previous data, it can be concluded that GLN-189 was the key amino acid residue interacting with the three structures (i.e., compounds **5**, **7**, and **8**) via H-bonding, while both the LEU-27 and MET-49 amino acid residues were the only residues hydrophobically interacting with them. The active steroids (**5**, **7**, and **8**) were interacting with three of the amino acids (GLN-189, LEU-27, and MET-49), which were included in the active site pockets of the SARS-CoV-2 M^pro^ [5], illustrating its possible activity. The dynamic behaviors of the three structures were convergent, where they were highly fluctuating over the course of the simulation, in comparison with the co-crystalized inhibitor (Figure 4). Their average RMSDs were 2.4 Å, 2.6 Å, and 1.3 Å for compounds **5**, **7**, and **8**, respectively (Figure 4). The calculated radius of the gyration profile of M^pro^ in complex with each compound was also consistent, with an average of ~ 18.4 Å (Figure 5). With regard to the interaction energy of the three structures over the course of the simulation, they showed stable interaction energy profiles (i.e., electrostatic and van der Waals interaction energies), with an average of ~-80 kcal/mol (Figure 6). Accordingly, the calculated MM-PBSA binding energy of each compound (Table 3) was found to be convergent with or even higher than that of the co-crystalized inhibitor ML188, indicating the potential inhibitory activity of these compounds against the viral M^pro^. Such structural information will be of great interest during the future development of novel M^pro^ inhibitors, depending on the scaffolds of these compounds (i.e., compounds **5**, **7**, and **8**).

With regard to the probable inhibitory activity SARS-CoV-2’s M^pro^ towards the isolated sesquiterpenes and sterols, docking, followed by MDS experiments, revealed that the polyhydroxylated sterols have good potential to bind with and to inhibit the enzyme’s catalytic activity. According to the dynamic binding mode analysis, the stability of such derivatives was achieved inside the enzyme’s active site through the formation of multiple H-bonds between the compounds’ hydroxyl groups and a number of hydrophilic amino acids. Hence, losing such essential hydroxyl groups or masking them via acetylation might lead to unstable binding (i.e., compounds **3**, **4**, and **6**). Previously, a number of polyhydroxylated triterpenes (e.g., ursolic and maslinic acids) were found to significantly inhibit SARS-CoV’s M^pro^ [48,49]. Hence, such steroidal or triterpenoidal scaffolds may be promising in the future development of novel M^pro^ inhibitors.

## 4. Materials and Methods

### 4.1. General Experimental Procedures

Optical rotations were recorded on a Jasco DIP-370 polarimeter. The 1D (^1^H and ^13^C) and 2D (HSQC, HMBC, ^1^H-^1^H COSY, and NOESY) NMR experiments were recorded on a Bruker DRX 600 NMR spectrometer (Bruker, Billerica, MA, USA). HR-ESI-MS and HR-FAB-MS were measured on an LC-MS-Q-TOF (Agilent Tokyo, Japan) and a JMS-700 mass spectrometer (JEOL, Tokyo, Japan), respectively. Coupling constants are expressed in Hz, and chemical shifts in *δ* (ppm). Chromatographic separations were performed using column chromatography on a Merck silica gel (70–230), and Medium-Pressure Liquid Chromatography (MPLC) (Büchi Reveleris^®^ Prep system, Flawil, Switzerland) was performed using a silica gel cartridge (40 μM, 12 g) and a C-18 cartridge (WP, 20 μM, 4 g) with a UV-ELSD detector. Thin-layer chromatography (TLC) was performed on glass pre-coated silica gel 60 F_254_ plates (Merck, Darmstadt, Germany) and reversed phase (RP-18 F_254_), which were visualized under UV light at (254 and 365 nm) and sprayed with 5% MeOH-H_2_SO_4_ reagent, followed by heating for 2–3 min.

### 4.2. Animal Material

*Heteroxenia fuscescens* soft coral was collected and identified by Dr. Aldoushy Mahdy (Faculty of Science, Al-Azhar University, Assiut branch, Egypt) from the Red Sea in front of the National Institute of Oceanography and Fisheries, Hurghada, Egypt. A voucher sample (HF 20) has been deposited at the Pharmacognosy Department, Faculty of Pharmacy, Al-Azhar University, Assiut Branch, Egypt.

### 4.3. Extraction and Isolation

The fresh material of *H. fuscescens* (370 g) was sliced and exhaustively extracted with 90% MeOH. The total MeOH extract (8.9 g) was fractionated using silica gel CC and eluted with 100% *n*-Hexane, *n*-Hexane:EtOAc, EtOAc, and MeOH (100%) to obtain five fractions of *n*-Hexane (1.6 g), *n*-Hexane:EtOAc (50:50, 1.4 g), EtOAc (100%, 2.3 g), EtOAc:MeOH (50:50, 2.7 g), and MeOH (100%, 4.2 g). The *n*-Hexane:EtOAc (50:50) (1.4 g) fraction was passed over a normal flash column to yield four sub-fractions (HE I-IV). Sub-fraction (HE-I) was separated on silica gel CC using *n*-Hexane:EtOAc in a gradient manner to afford pure compound **2** (2.0 mg) and compound **3** (11.3 mg), which were finally purified via NPTLC using *n*-Hexane:EtOAc (40:60). Sub-fraction HE-II was separated by RP-18 MPLC using a MeOH-H_2_O gradient from (30:70) to 100% MeOH to afford compound **1** (1.2 mg). Sub-fraction HE-IV was separated on NPTLC using *n*-Hexane:EtOAc (30:70) to obtain compound **4** (6.3 mg). The EtOAc fraction (2.3 g) was subjected to an RP-18 flash column using MeOH-H_2_O in a gradient manner (50:50 to 80:20) to afford six sub-fractions (Et I–VI). Compound **8** (2.2 mg) was obtained from sub-fraction Et II by using RP_18_-TLC and using (MeOH: H_2_O, 70:30) as a solvent system. Sub-fraction Et IV was passed over Sephadex (LH-20) and eluted with MeOH 100% to obtain compound **5** (10.2 mg) and compound **7** (3.1 mg). Compound **6** (4.2 mg) was obtained from sub-fraction VI, which passed over silica gel CC and was eluted with *n*-Hexane:EtOAc (85:15) to EtOAc:MeOH (90:10).

### 4.4. Spectral Data

#### 4.4.1. Fusceterpene A (**1**)

Colorless oil; [a] _D_^20^ + 62.2 (c 0.001, MeOH); for ^1^H and ^13^C-NMR data, see (Table 1). Positive HR-FAB-MS 235.1701 [M+H]^+^ (calcd for C_15_H_23_O_2_, 235.1698 *m/z*).

#### 4.4.2. Fuscesterol A (**4**)

White amorphous powder; [a] _D_^20^ + 25.4 (c 0.002, MeOH); for ^1^H and ^13^C-NMR data, see (Table 2). Positive HR-ESI-MS at *m/z* 527.3673 [M+Na]^+^ (calcd for C_31_H_52_NaO_5,_ 527.3712 *m/z*).

### 4.5. Molecular Docking and Molecular Dynamic Simulation

(See Appendix A).

## 5. Conclusions

This study highlighted the significance of identifying antiviral secondary metabolites from the Red Sea’s soft corals as naturally occurring SARS-CoV-2 inhibitors. A new steroid and a new sesquiterpene, along with six known compounds, were effectively isolated and structurally characterized, and they were essentially evaluated via in silico study against SARS-CoV-2 proteins. Through a stable hydrogen bond, the purified steroids containing a (3*β*,5*α*,6*β*-trihydroxy) moiety interact with the amino acid residue GLU-189, which highlights it as a perfect framework for the future creation of SARS-CoV-2 M^pro^ inhibitory medications.

## Figures and Tables

**Figure 1 molecules-27-07369-f001:**
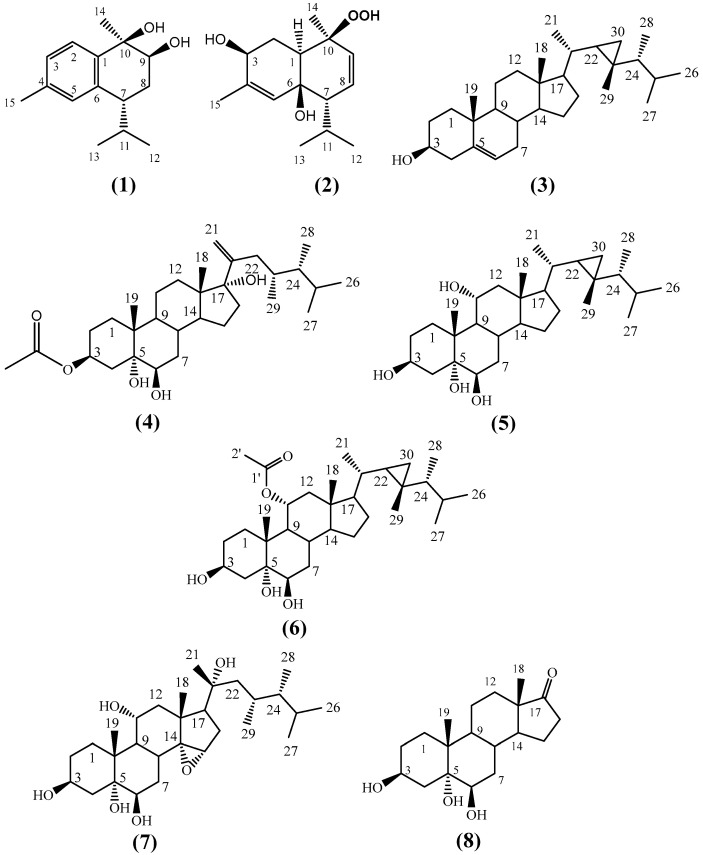
Structure of the isolated compounds (**1**)–(**8**) from *H. fuscescens*.

**Figure 2 molecules-27-07369-f002:**
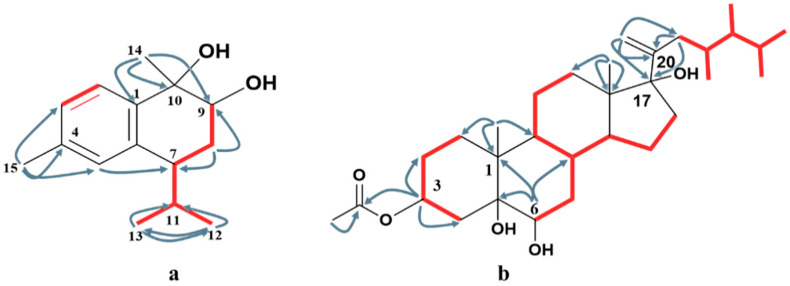
Key HMBC (

) and ^1^H-^1^H COSY (

) correlations of compounds **1** (**a**) and **4** (**b**).

**Figure 3 molecules-27-07369-f003:**
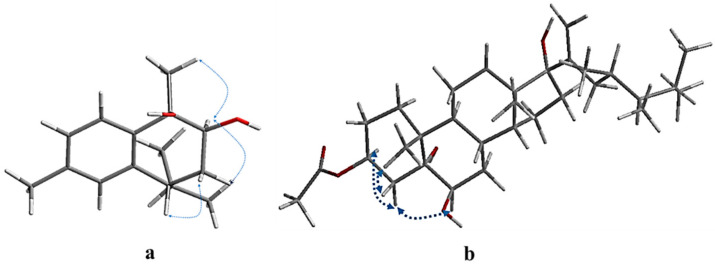
Key NOESY correlations of compounds **1** (**a**) and **4** (**b**).

**Figure 4 molecules-27-07369-f004:**
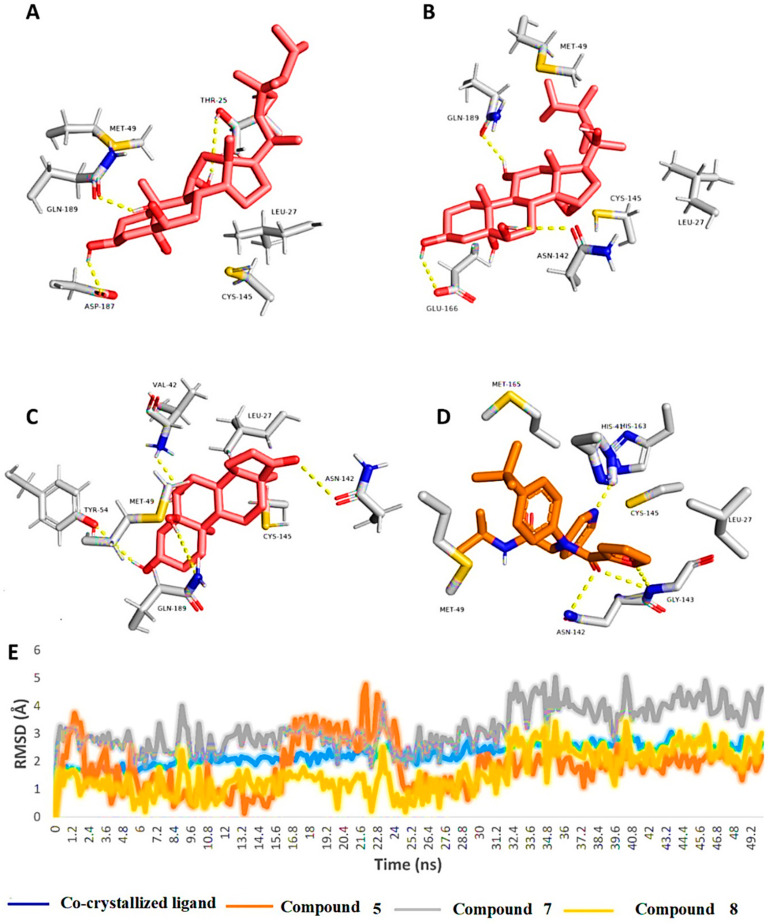
Binding modes of compounds **5**, **7**, and **8** along with the co-crystalized inhibitor ML188 (**A**–**D**, respectively) inside the active site of SARS-CoV-2′s M^pro^ (PDB code: 7L0D). These binding modes were extracted from the 50-ns-long MDS trajectories as the most populated poses. (**E**): RMSDs of compounds **5**, **7**, and **8** along with the co-crystalized inhibitor ML188 inside the active site of SARS-CoV-2’s M^pro^ over the course of 50-ns-long MDS.

**Figure 5 molecules-27-07369-f005:**
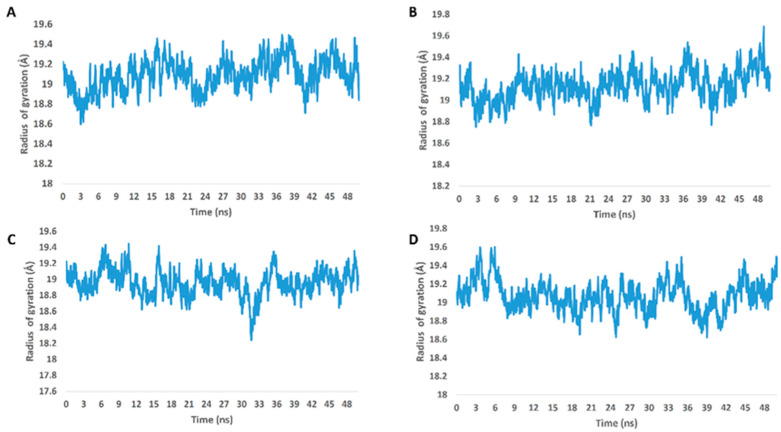
Radius of gyration of M^pro^ in complex with compounds **5**, **7**, and **8** along with the co-crystalized inhibitor ML188 (**A**–**D**, respectively) over the course of 50-ns-long MDS.

**Figure 6 molecules-27-07369-f006:**
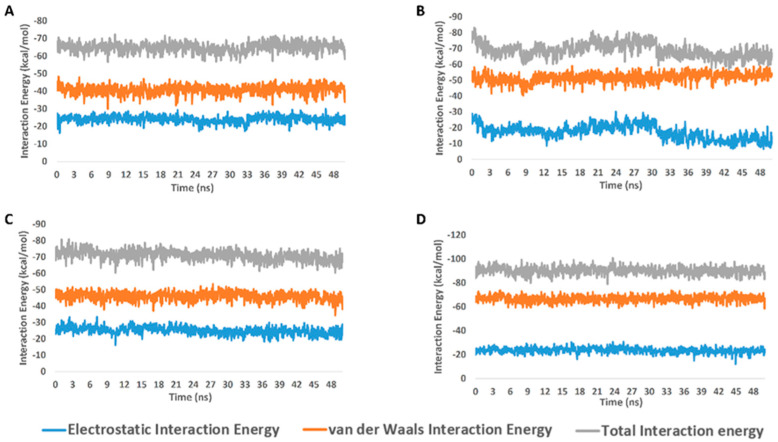
Interaction energy profiles of compounds **5**, **7**, and **8** along with the co-crystalized inhibitor ML188 in complex with M^pro^ (**A**–**D**, respectively) over the course of 50-ns-long MDS.

**Table 1 molecules-27-07369-t001:** ^1^H and ^13^C NMR data of compound **1**.

Position	*δ_H_* ^a^			*δ_C_* ^a^
	Chemical Shift (ppm)	Multiplicity	[*J* in (Hz)]	
1	---	---	---	139.5
2	7.48	d	7.9	125.4
3	7.06	d	7.9	127.5
4	---	---	---	137.7
5	7.03	s	---	128.8
6	---	---	---	136.8
7	2.84	m	---	41.5
8	2.01 and 1.85	m m	--- ---	27.3
9	4.01	dd	3.9, 11.3	73.8
10	---	---	---	74.5
11	2.20	m	---	33.6
12	1.06	d	7.0	21.6
13	0.78	d	7.0	18.01
14	1.41	s	---	24.31
15	2.30	s	---	21.2

^a^ Measured in CDCl_3_, ^1^H-NMR (600 MHz) and ^13^C-NMR (150 MHz).

**Table 2 molecules-27-07369-t002:** ^1^H and ^13^C NMR data of compound **4**.

Position	*δ_H_* ^a^			*δ_C_* ^a^
	Chemical Shift (ppm)	Multiplicity	[*J* in (Hz)]	
1	1.67 and 1.32	m m	--- ---	33.3
2	1.79 and 1.56	m m	--- ---	27.9
3	5.17	m	---	73.0
4	2.16 and 1.58 *	m ---	--- ---	38.0
5	---	---	---	76.6
6	3.46	t	3.0	76.4
7	1.75 and 1.53	m m	--- ---	35.2
8	1.73	m	---	32.1
9	1.43	m	---	46.4
10	---	---	---	39.5
11	1.44	m	---	22.1
12	1.72 * and 1.37	--- dt	--- 3.6, 12.6	32.2
13	---	---	---	49.9
14	1.9	m	---	51.0
15	1.70 and 1.23	m m	--- ---	24.3
16	2.35 and 1.57*	td ---	3.0, 14.4 ---	35.8
17	---	---	---	87.7
18	0.65	s	---	16.3
19	1.18	s	---	17.2
20	---	---	---	152.3
21	5.14 and 4.89	br.s br.s	--- ---	112.0
22	2.02	m	---	41.5
23	1.93	m	---	33.8
24	1.12	m	---	45.3
25	1.65	m	---	32.0
26	0.93	d	6.6	20.7
27	0.89	d	6.6	21.4
28	0.80	d	6.6	11.8
29 1′ 2′	0.78	d	6.6 --- ---	14.0 172.8 21.9
	---	---		
	1.98	s		

^a^ Measured in CD_3_OD, ^1^H-NMR (600 MHz) and ^13^C-NMR (150 MHz). * Multiplicity not determined due to overlapping.

**Table 3 molecules-27-07369-t003:** MM-PBSA binding free energies of compounds **5**, **7**, and **8** along with the co-crystalized inhibitor ML188 in complex with M^pro^.

Energy Component	Compound 5	Compound 7	Compound 8	Co-Crystalized Ligand
ΔG_gas_	−33.828	−32.723	−31.746	−41.9382
ΔG_solv_	17.1182	18.9182	19.8337	27.6434
ΔG_Total_	−16.7094	−13.8044	−11.9126	−14.2948

## Data Availability

Not applicable.

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
