# Peer review of "A Comprehensive In Silico Study of New Metabolites from Heteroxenia fuscescens with SARS-CoV-2 Inhibitory Activity"

_molecules, 2022, doi:10.3390/molecules27217369_

Round 1
Reviewer 1 Report
1. What was the rationale of this study? The authors need to justify as thousands of literatures are published in lieu of COVID19 therapeutics. Moreover, many in silico studies are reporting various compounds acting as inhibitors of one or more protein of SARS-CoV-2. But, without further validation, how are these studies vital?
2. Consequently, MPro and CDKs inhibitors are considered as an attractive target for drugs used to treat SARS-COV-2 virus. The Introduction section is redundant without much information about the SARS-CoV-2. It should be rewritten focusing on the important proteins of it and also highlighting the importance of these proteins as therapeutic target.
These literatures will aid the authors.
https://doi.org/10.1042/BSR20201256
https://doi.org/10.1016/j.ijbiomac.2021.02.071
3. have considerable docking scores (< -8.0 kcal/mol). Is it correctly written?
A table should be made depicting the docking score of the all the compounds with all the viral proteins.
4. The Subsequent MDS validation experiments (50 ns long). 50 ns simulations are not enough. The authors should have performed at least 100 ns simulation to decipher structural dynamics.
5. Only RMSD was investigated. What about other parameters. The MD analysis is incomplete and irrelevant. It should be performed extensively or else just for sake of adding an additional in silico tool is of no use.
6. The overall manuscript needs considerable revision of English language as it is full of grammatical flaws.

Author Response
Manuscript ID: molecules-1952247
Type of manuscript: Article
Title: A comprehensive in silico study of new metabolites from Heteroxenia fuscescens with SARS-CoV-2 Inhibitory Activity.
Thank you very much for your comments and recommendations, the requested comments are already included as track changes in the main manuscript, here, the responds to each reviewer comments separately.
Reviewer Reports:
Reviewer 1
1- What was the rationale of this study? The authors need to justify as thousands of literatures are published in lieu of COVID19 therapeutics. Moreover, many in silico studies are reporting various compounds acting as inhibitors of one or more protein of SARS-CoV-2. But, without further validation, how are these studies vital?
Response:
Thank you very much for your comments. This study recorded the isolation of two new compounds, one of them sesquiterpenoid and the second is a steroid in nature. Additionally, this is the first report for in silico study of the isolated compounds as SARS-CoV-2 inhibitors. Based on the results of this research, steroids with (3β,5α,6β-trihydroxy) steroids is a promising candidate for the future development.
2- Consequently, MPro and CDKs inhibitors are considered as an attractive target for drugs used to treat SARS-COV-2 virus. The Introduction section is redundant without much information about the SARS-CoV-2. It should be rewritten focusing on the important proteins of it and also highlighting the importance of these proteins as therapeutic target.
These literatures will aid the authors.
https://doi.org/10.1042/BSR20201256
https://doi.org/10.1016/j.ijbiomac.2021.02.071.
Response:
Thank you very much for your comments and suggestion, we improved the introduction part and covered all mentioned points.
3- Have considerable docking scores (< -8.0 kcal/mol). Is it correctly written?
Response:
Thank you very much for your comment, yes, this value was chosed according to the docking score of the co-crystalized ligand (-8.13 kcal/mol), so scores less than this value will be interesting and hence, they were chosen.
A table should be made depicting the docking score of the all the compounds with all the viral proteins.
Response:
Thank you very much for your comments, a new table with the required docking scores was added in the revised supplementary file.
4- The Subsequent MDS validation experiments (50 ns long). 50 ns simulations are not enough. The authors should have performed at least 100 ns simulation to decipher structural dynamics.
Response:
We thank the reviewer for this comment. Actually, we found that the RMSDs of the simulated compounds reached a steady state after ⁓ 30 ns (Figure 4). So, we believed that a run time of about 50 ns will be a good representative for the binding dynamics of the selected compounds, particularly, this kind of computations is time and resources consuming.
5- Only RMSD was investigated. What about other parameters. The MD analysis is incomplete and irrelevant. It should be performed extensively or else just for sake of adding an additional in silico tool is of no use.
Response:
We thank the reviewer for this comment. In addition to the simulations and the parameters extracted from them (either the RMSD or absolute binding free energy), we added further analysis e.g. radius of gyration and interaction energy profiles over the course of simulation to indicate the potential affinity and binding stability of compounds 5, 7, and 8 with Mpro. Moreover, we added the calculated MM-PBSA binding energies of each compound with Mpro.
6- The overall manuscript needs considerable revision of English language as it is full of grammatical flaws.
Response:
Thank you very much of your comments, we corrected and improved the English language of our manuscript.
Reviewer 2 Report
Title: A Comprehensive In silico Study of New Metabolites from Heteroxenia fuscescens with SARS-CoV-2 Inhibitory Activity
Journal: Molecules
The topic is of interest, and the manuscript is well illustrated.
Major Comments:
1. Are there controversies in this field? What are the most recent and important achievements in the field? In my opinion, answers to these questions should be emphasized. Perhaps, in some cases, novelty of the recent achievements should be highlighted by indicating the year of publication in the text of the manuscript.
2. The results and discussion section is very weak and no emphasis is given on the discussion of the results like why certain effects are coming in to existence and what could be the possible reason behind them?
3. Discussion should be separated from results.
4. Results and conclusion: The section devoted to the explanation of the results suffers from the same problems revealed so far. Your storyline in the results section (and conclusion) is hard to follow. Moreover, the conclusions reached are really far from what one can infer from the empirical results.
5. The discussion should be rather organized around arguments avoiding simply describing details without providing much meaning. A real discussion should also link the findings of the study to theory and/or literature.
6. Spacing, punctuation marks, grammar, and spelling errors should be reviewed thoroughly. I found so many typos throughout the manuscript.
7. English is modest. Therefore, the authors need to improve their writing style. In addition, the whole manuscript needs to be checked by native English speakers.
8. Why Heteroxenia fuscescens? Need to justify.
9. Scientific names should be italics.
10. Conclusions need to rewrite completely.
Author Response
Manuscript ID: molecules-1952247
Type of manuscript: Article
Title: A comprehensive in silico study of new metabolites from Heteroxenia fuscescens with SARS-CoV-2 Inhibitory Activity.
Thank you very much for your comments and recommendations, the requested comments are already included as track changes in the main manuscript, here, the responds to each reviewer comments separately.
Reviewer Reports:
Reviewer 2
Title: A Comprehensive In silico Study of New Metabolites from Heteroxenia fuscescens with SARS-CoV-2 Inhibitory Activity
Journal: Molecules
The topic is of interest, and the manuscript is well illustrated.
Major Comments:
1- Are there controversies in this field? What are the most recent and important achievements in the field? In my opinion, answers to these questions should be emphasized. Perhaps, in some cases, novelty of the recent achievements should be highlighted by indicating the year of publication in the text of the manuscript.
Response:
Thank you very much for your comments and suggestions. We improved the introduction part to include the recent achievements in this field.
2- The results and discussion section is very weak and no emphasis is given on the discussion of the results like why certain effects are coming in to existence and what could be the possible reason behind them?
Response:
Thank you very much for your comment. The link between the binding of compounds 5, 7, and 8 and their potential as Mpro inhibitor was further discussed in the revised manuscript.
3- Discussion should be separated from results.
Response:
Thank you very much for your comment. We fully understand your point of view, but the two parts are tangled with each other and difficult to separate. Additionally, the result and discussion parts are usually put in one section in molecules journal papers.
4- Results and conclusion: The section devoted to the explanation of the results suffers from the same problems revealed so far. Your storyline in the results section (and conclusion) is hard to follow. Moreover, the conclusions reached are really far from what one can infer from the empirical results.
Response:
Thank you very much for your comment; we improved the conclusion and result parts to cover all points mentioned by the reviewer.
5- The discussion should be rather organized around arguments avoiding simply describing details without providing much meaning. A real discussion should also link the findings of the study to theory and/or literature.
Response:
Thank you very much for your comment and suggestion. We reconstructed the discussion part to link the findings to theory and published literature.
6- Spacing, punctuation marks, grammar, and spelling errors should be reviewed thoroughly. I found so many typos throughout the manuscript.
Response:
Thank you very much for you comment, we checked and corrected the typos and errors throughout the manuscript.
7- English is modest. Therefore, the authors need to improve their writing style. In addition, the whole manuscript needs to be checked by native English speakers.
Response:
Thank you very much of your comments, we corrected and improved the English language of our manuscript.
8- Why Heteroxenia fuscescens? Need to justify.
Response:
Thank you very much for you comment, we illustrated this point in introduction part of our manuscript.
9- Scientific names should be italics.
Response:
Thank you very much, we revised the scientific names.
10- Conclusions need to rewrite completely.
Response:
Thank you very much for the comment. We improved and reconstructed the conclusion to sum all the important outcomes of our manuscript.
Reviewer 3 Report
The authors report the results from the chemical investigation of the total extract of the Egyptian soft coral Heteroxenia fuscescens. They have successfully isolated 8 compounds two of which are new - sesquiterpene fusceterpene A and a sterol fuscesterol A. The compounds were fully and correctly identified using NMR and HRMS techniques and compare the known ones with the literature data.
The research continues with in silico analysis of the compounds. This study pointed out the importance of the isolation of bioactive metabolites from the Red Sea marine organisms as a source of naturally occurring compounds in alleviating SARS-CoV-2 infection.
Despite the interesting work, I have a few questions, remarks, and recommendations:
Please check and correct your manuscript for grammar, punctuation, and spelling errors!
At 3.3 Extraction and isolation
-Please use the full name of the solvent n-Hexane instead of n-Hex.
-Use fraction instead of fr.
Make sure you unify g or g. or gm
Line 243 – missing g after 2.7.
Please read carefully and correct the procedure.
What does it mean amorphous powder? If is powder is powder if is oil is oil. Are you sure that the compound was fully purified? Have you tried to crystalize the isolated compound?
Please provide for the two new compounds (Fusceterpene A and Fuscesterol A) full spectral data characteristics, including 1H-NMR, and 13C-NMR in the material and method section before the conclusion, instead of giving a reference – “1H and 13C-NMR data see 262 (Table 2)”.
Molecular docking and molecular dynamic simulation experimental data have to be included in the article instead of the supplementary file.
Please calculate the mass error for HRMS in ppm.
The conclusion is very short and does not provide sufficient a conclusive resume of the obtained results. Please update the resume and describe the positives found out in the current research.
I suggest reconsidering the article for publication after a major revision.
Author Response
Manuscript ID: molecules-1952247
Type of manuscript: Article
Title: A comprehensive in silico study of new metabolites from Heteroxenia fuscescens with SARS-CoV-2 Inhibitory Activity.
Thank you very much for your comments and recommendations, the requested comments are already included as track changes in the main manuscript, here, the responds to each reviewer comments separately.
Reviewer Reports:
Reviewer 3
The authors report the results from the chemical investigation of the total extract of the Egyptian soft coral Heteroxenia fuscescens. They have successfully isolated 8 compounds two of which are new - sesquiterpene fusceterpene A and a sterol fuscesterol A. The compounds were fully and correctly identified using NMR and HRMS techniques and compare the known ones with the literature data.
The research continues with in silico analysis of the compounds. This study pointed out the importance of the isolation of bioactive metabolites from the Red Sea marine organisms as a source of naturally occurring compounds in alleviating SARS-CoV-2 infection.
Despite the interesting work, I have a few questions, remarks, and recommendations:
Please check and correct your manuscript for grammar, punctuation, and spelling errors!
Response:
Thank you very much for the comment. We checked and corrected the typos and errors throughout the manuscript.
At 3.3 Extraction and isolation
-Please use the full name of the solvent n-Hexane instead of n-Hex.
-Use fraction instead of fr.
Make sure you unify g or g. or gm
Line 243 – missing g after 2.7.
Please read carefully and correct the procedure.
Response:
Thank you very much for the comment. The full name of the solvent n-Hexane and the word fraction were modified. We unified the symbol (g). The missing g added after 2.7 in lone 243. We also went through the procedure and corrected it.
What does it mean amorphous powder? If is powder is powder if is oil is oil. Are you sure that the compound was fully purified? Have you tried to crystalize the isolated compound?
Response:
Thank you very much for the comment. Amorphous powder means that it is non-crystalline material that possesses no long-range order. Yes, this compound is fully purified. We tried to crystalize the new isolated compounds with pure and\or mixed solvents, but without success. So, we mentioned them as “amorphous powder”.
Please provide for the two new compounds (Fusceterpene A and Fuscesterol A) full spectral data characteristics, including 1H-NMR, and 13C-NMR in the material and method section before the conclusion, instead of giving a reference – “1H and 13C-NMR data see 262 (Table 2)”.
Response:
Thank you very much for the comment. The full data of the new compounds (Fusceterpene A and Fuscesterol A) are listed in Tables 1 and 2. Providing them again in the material and method section may be a repetition of the same data.
Molecular docking and molecular dynamic simulation experimental data have to be included in the article instead of the supplementary file.
Response:
Thank you very much for the comment. We included the important Molecular docking and molecular dynamic simulation results in the main manuscript, and moved the experimental section to the supplementary file to avoid over length of the manuscript and to be easy for the reader to follow.
Please calculate the mass error for HRMS in ppm.
Response:
Thank you very much for the comment. The mass error of fusceterpene A (1) is 1.3 ppm, and that of fuscesterol A (4) is 7.4 ppm.
The conclusion is very short and does not provide sufficient a conclusive resume of the obtained results. Please update the resume and describe the positives found out in the current research.
Response:
Thank you very much for the comment and suggestion. We improved and reconstructed the conclusion to sum all the important outcomes of our manuscript.
I suggest reconsidering the article for publication after a major revision.
Response:
Thank you very much for your valuable comments and suggestions.
Round 2
Reviewer 1 Report
All the comments have been addressed in an appropriate manner and the manuscript can now be accepted for publication.
Author Response
Manuscript ID: molecules-1952247
Type of manuscript: Article
Title: A comprehensive in silico study of new metabolites from Heteroxenia fuscescens with SARS-CoV-2 Inhibitory Activity.
Thank you very much for your comments and recommendations, the requested comments are already included as track changes in the main manuscript, here, the responds to each reviewer comments separately.
Reviewer Reports:
Reviewer 1
All the comments have been addressed in an appropriate manner and the manuscript can now be accepted for publication.
Response:
Thank you very much for your reviewing our manuscript.

Reviewer 2 Report
Title: A Comprehensive In silico Study of New Metabolites from Heteroxenia fuscescens with SARS-CoV-2 Inhibitory Activity
Journal: Molecules
Major Comments:
1. The results and discussion section are still weak, and no emphasis is given on the discussion of the results like why certain effects are coming into existence and what could be the possible reason behind them.
2. Discussion should be separated from results.
3. Molecular docking and molecular dynamic simulation (See supplementary material). Need to add some insights.
4. The discussion should be rather organized around arguments avoiding simply describing details without providing much meaning. A real discussion should also link the findings of the study to theory and/or literature.
5. English is modest. Therefore, the authors need to improve their writing style. In addition, the whole manuscript needs to be checked by native English speakers.
6. Conclusions need to rewrite completely.
Author Response
Manuscript ID: molecules-1952247
Type of manuscript: Article
Title: A comprehensive in silico study of new metabolites from Heteroxenia fuscescens with SARS-CoV-2 Inhibitory Activity.
Thank you very much for your comments and recommendations, the requested comments are already included as track changes in the main manuscript, here, the responds to each reviewer comments separately.
Reviewer Reports:
Reviewer 2
1- The results and discussion section are still weak, and no emphasis is given on the discussion of the results like why certain effects are coming into existence and what could be the possible reason behind them.
Response:
Thank you very much for your comments. We added a new discussion part in the revised version showing the potential of this class of compounds as a promising scaffold for the future development of new Mpro inhibitors, and how our in silico study explained the structural features needed for probable good binding inside the enzyme’s active site.
2- Discussion should be separated from results.
Response:
Thank you very much for your suggestion. We separated the discussion and result parts from each other in the revised manuscript.
3- Molecular docking and molecular dynamic simulation (See supplementary material). Need to add some insights.
Response:
Thank you very much for your comments. New details regarding the methods of docking and molecular dynamics simulation were added in the supplementary material.
4- The discussion should be rather organized around arguments avoiding simply describing details without providing much meaning. A real discussion should also link the findings of the study to theory and/or literature.
Response:
Thank you very much for your comments. We added a new discussion part in the revised version showing the potential of this class of compounds as a promising scaffold for the future development of new Mpro inhibitors, and how our in silico analysis explained the structural features needed for probable good binding inside the enzyme’s active site.
5- English is modest. Therefore, the authors need to improve their writing style. In addition, the whole manuscript needs to be checked by native English speakers.
Response:
Thank you very much for your comments. Our revised manuscript was subjected to English editing by native English speaker.
6- Conclusions need to rewrite completely.
Response:
Thank you very much for your comment. The conclusion section was rewritten as required in the revised manuscript.

Reviewer 3 Report
Dear Authors,
Now I confirm that your manuscript looks much better.
I give my positive consent for the presented manuscript to be published.
Kind regards,
Author Response
Manuscript ID: molecules-1952247
Type of manuscript: Article
Title: A comprehensive in silico study of new metabolites from Heteroxenia fuscescens with SARS-CoV-2 Inhibitory Activity.
Thank you very much for your comments and recommendations, the requested comments are already included as track changes in the main manuscript, here, the responds to each reviewer comments separately.
Reviewer Reports:
Reviewer 3
Dear Authors,
Now I confirm that your manuscript looks much better.
I give my positive consent for the presented manuscript to be published.
Kind regards,
Response:
Thank you very much for your reviewing our manuscript.
